# MiR-199a-3p Induces Mesenchymal to Epithelial Transition of Keratinocytes by Targeting RAP2B

**DOI:** 10.3390/ijms232315401

**Published:** 2022-12-06

**Authors:** Moamen Masalha, Tal Meningher, Adi Mizrahi, Aviv Barzilai, Hilla Tabibian-Keissar, Devorah Gur-Wahnon, Iddo Z. Ben-Dov, Joshua Kapenhas, Jasmine Jacob-Hirsch, Raya Leibowitz, Yechezkel Sidi, Dror Avni

**Affiliations:** 1Laboratory of Molecular Cell Biology, Center for Cancer Research, Department of Medicine C, Sheba Medical Center, Tel Hashomer 52621, Israel; 2Faculty of Medicine, Sackler School of Medicine, Tel Aviv University, Tel Aviv 69978, Israel; 3Department of Dermatology, Institute of Pathology Sheba Medical Center, Tel Hashomer 52621, Israel; 4Department of Pathology, Sheba Medical Center, Tel Hashomer 52621, Israel; 5Laboratory of Medical Transcriptomics, Nephrology and Hypertension Services, Hadassah-Hebrew University Medical Center, Jerusalem 91120, Israel; 6Center for Cancer Research Sheba Medical Center, Tel Hashomer 52621, Israel; 7Oncology institute, Shamir Medical Center, Zerifin 70300, Israel

**Keywords:** cutaneous squamous cell carcinoma (CSCC), miR-199a-3p, epithelial-mesenchymal transition (EMT), RAP2B

## Abstract

Cutaneous squamous cell carcinoma (CSCC) is an epidermal skin cancer that evolves from normal epidermis along several pre-malignant stages. Previously we found specific miRNAs alterations in each step along these stages. miR-199a-3p expression decreases at the transition to later stages. A crucial step for epithelial carcinoma cells to acquire invasive capacity is the disruption of cell–cell contacts and the gain of mesenchymal motile phenotype, a process known as epithelial-to-mesenchymal transition (EMT). This study aims to study the role of decreased expression of miR-199a-3p in keratinocytes’ EMT towards carcinogenesis. First, we measured miR-199a-3p in different stages of epidermal carcinogenesis. Then, we applied Photoactivatable Ribonucleoside-Enhanced Crosslinking and Immunoprecipitation (PAR-CLIP) assay to search for possible biochemical targets of miR-199a-3p and verified that Ras-associated protein B2 (RAP2B) is a bona-fide target of miR-199a-3p. Next, we analyzed RAP2B expression, in CSCC biopsies. Last, we evaluated possible mechanisms leading to decreased miR-199a-3p expression. miR-199a-3p induces a mesenchymal to epithelial transition (MET) in CSSC cells. Many of the under-expressed genes in CSCC overexpressing miR-199a-3p, are possible targets of miR-199a-3p and play roles in EMT. RAP2B is a biochemical target of miR-199a-3p. Overexpression of miR-199a-3p in CSCC results in decreased phosphorylated focal adhesion kinase (FAK). In addition, inhibiting FAK phosphorylation inhibits EMT marker genes’ expression. In addition, we proved that DNA methylation is part of the mechanism by which miR-199a-3p expression is inhibited. However, it is not by the methylation of miR-199a putative promoter. These findings suggest that miR-199a-3p inhibits the EMT process by targeting RAP2B. Inhibitors of RAP2B or FAK may be effective therapeutic agents for CSCC.

## 1. Introduction

Skin cancer is one of the most common human malignancies worldwide. The three major forms of skin cancer are basal cell carcinoma, cutaneous squamous cell carcinoma (CSCC), and melanoma [1]. The development and progression of CSCC are strongly associated with chronic exposure to sunlight ultraviolet radiation and thus it is preferentially located on sun-exposed skin parts of the body [2,3]. The process underlying the transformation of a healthy epidermis into CSCC, is a multi-step process with several pathologically defined stages [4,5]. Healthy skin evolves to solar-elastosis (SE) and then to actinic-keratosis (AK) with varying degrees of keratinocyte (KC) dysplasia, termed keratinocytic intraepidermal neoplasia (KIN). KIN degree is defined according to the depth of involvement of dysplastic KC in the epidermis. The most advanced stage is CSCC [4,5,6,7].

miRNAs are small, noncoding, ~22 nucleotide RNA molecules that function as regulators of gene expression [8,9,10]. miRNAs are involved in many biological processes. miRNAs affect cancer development, progression, and metastasis by altering the expression of both oncogenes and tumor suppressor genes [8,10,11]. Tumorigenesis is facilitated by the loss of certain miRNAs. For example, miR-15/16 cluster in chronic lymphocytic leukemia [12], miR-34a in uveal melanoma [13], and miR-31 in mesothelioma [14]. We found that the expression of a large cluster of miRNAs on human chromosome 14q32 is silenced in melanoma [15]. In contrast, cancer advances by overexpression of miR-17~92 cluster members, which promotes migration and invasion in several malignancies [16,17,18,19].

Recently we showed that there are major alterations in miRNAs expression along the malignant evolution of KC [20]. Moreover, we could define specific miRNAs change at specific stages of this transformation [20]. One of those miRNAs is miR-199a-3p, whose expression decreases at the transition from AK-KIN1/2 to AK-KIN3 [20]. MiR-199a-3p can be transcribed from two genes: *miR-199a-1* at chromosome 19p13.2, in one of the introns of the dynamin2 (*DMN2*) gene and *miR-199a-2* at chromosome 1q24.3, within the long non-coding-RNA gene dynamin3 opposite strain (*DNM3OS*). The main transcript from both genes is miR-199a-3p.

In our previous work, miR-199a-3p expression was one of the highest-expressed miRNAs in healthy skin biopsies [20]. MiR-199a-3p expression decreases in hepatocellular carcinomas [21], prostate cancer [22], endometrial cancer [23], CSCC [24], and in testicular malignancy [25,26]. In contrast, increased expression of miR-199-3p was found in gastric cancer [27].

## 2. Results

### 2.1. miR-199a-3p Expression Is Decreased at the Advanced Stages of KC Malignant Evolution

Total RNA was extracted from FFPE biopsies of healthy epidermis, severe SE, AK-KIN1/2, AK-KIN3, and well-differentiated CSCC, and subjected to qRT-PCR with miR-199-3p specific primers. As shown in Figure 1A, there is a significant decrease in miR-199a-3p expression along the normal to CSCC axis, most strikingly in the AK-KIN1/2 to AK-KIN3 transition.

To study the biological effects of miR-199a-3p on CSCC, we first analyzed the expression of miR-199a-3p in different CSCC cell-lines in comparison to healthy primary human KC cells (PHK). Shown in Figure 1B, all CSCC cell-lines express much less miR-199a-3p compared to PHK.

### 2.2. miR-199a-3p Expression Is Not Correlated with the Expression of DNM3OS or DNM2

In the human genome, miR-199a-3p is encoded from *miR-199a-1* in chromosome-19 within an intron of the *DNM2* gene, and from *miR-199a-2* in chromosome-1 within an intron of the *DNM3OS* gene. One possibility for the decrease of miR-199a-3p expression in CSCC is decreased expression of these two genes. Hence, we compared the expression levels of these two genes between PHK to CSCC cell-lines, and between tumor or healthy skin biopsies. We found that the expression of *DNM3OS*, a non-coding RNA, expression increases in CSCC biopsies compared to healthy skin biopsies (Figure 1C). Likewise, in CSCC cell-lines, *DNM3OS* also increases compared to PHK but not statistically significantly (Figure 1D). We did not find any difference in the expression of *DNM2* between tumor to healthy biopsies (Figure 1E). *DNM2* decreased in CSCC cell-lines compared to PHK (Figure 1F), but not to the same extent as miR-199a-3p (Figure 1B). The above results suggest that the decreased expression of miR-199a-3p in CSCC is not because of the decreased expression of these two genes.

### 2.3. The Role of DNA Methylation in miR-199a-3p Expression

To analyze whether methylation plays a role in the silencing of miR-199a-3p in CSCC, six different CSCC cell-lines were treated with 5-AZA-2′deoxycytidine (5-AZA) continuously for five days. Shown in Figure 2A, in all cells there was a significant increase in miR-199a-3p expression compared to untreated cells.

Although the expression of miR-199a-3p seems to be regulated by methylation, both the expression of DNM2 and DNM3OS were not affected by 5-AZA treatment (Figure 2B). Reinforcing the conclusion that the expression regulation of miR-199a-3p is detached from the expression of both *DNM2* and *DNM3OS*.

### 2.4. miR-199a-1 and miR-199a-2 Putative Promoters Methylation

From the 5-AZA results, we expected that both or at least one of miR-199a-1 or miR-199a-2 promoters will be unmethylated in PHK and healthy skin biopsies, and methylated in CSCC lines and biopsies. However, it seems that the miR-199a-1 putative promoter region is methylated in all samples (Figure 2C). Remarkably, the miR-199a-2 putative promoter region was found to be un-methylated in three different CSCC biopsies. In contrast, in most CSCC cell-lines, this region is methylated. Moreover, in PHK and in healthy skin biopsies, this region is methylated (Figure 2C).

### 2.5. Effects of miR-199a-3p in CSCC Cells

To determine the functional role of miR-199a-3p in CSCC, we generated CSCC cell-lines, stably expressing the pre-miR-199a-3p or a control vector (Figure 3A). We observed a change in the cellular morphology; miR-199a-3p transfected cells acquired epithelial morphology, in contrast to the typical spindle-shaped mesenchymal cell morphology of control cells (Figure 3B). This alteration is suggestive of mesenchymal to epithelial transition (MET) activated by miR-199a. Using the xCELLigence real-time-cell analyzer system, we found that miR-199a effect CSCC migration and proliferation. Whereas control-transfected cells, exhibited high proliferation and migration ability, miR-199a transfected cells displayed a significant reduction in both proliferation and migration (Figure 3C,D).

### 2.6. Effect of miR-199a-3p on MET/EMT Molecular Markers

To confirm that indeed the morphological changes we observed are MET, we compare the expression of known EMT/MET markers in MET1 and SCL-II overexpressing miR-199a to control MET1 and SCL-II, by qRT-PCR (Figure 3E,F) and Western blot (WB) analysis (Figure 3G). Shown in Figure 3 mRNAs of *SMAD3*, *SMAD4*, and *SERPINE1* decrease significantly in both cells overexpressing miR-199a-3p. MiR-199a-3p expression in MET1 cells results in a significant decrease in VIM expression, while SCL-II does not express VIM (Figure 3G). In SCL-II overexpressing miR-199a-3p, there is a significant decrease in the expression of *COL4A1*, *ID1*, and *ID2*. The WB analysis reinforces the mRNAs’ findings, as the protein levels of several other EMT markers; β-Catenin, Slug, and N-Cadherin decreased in cells overexpressing miR-199a-3p (Figure 3G).

### 2.7. Targets of miR-199-3p

To understand the molecular mechanism by which miR-199a-3p contributes to the MET phenotype, we searched for its potential targets using two different experimental systems. First, we analyzed an mRNA expression array of cells overexpressing miR-199a-3p and compared it to the matching control cells. Cluster analysis reveals that cells overexpressing miR-199a-3p are more closely clustered together than each of the lines is to its matching control cell (Appendix A). Using a cut-off of 2-fold change in expression, the expression of 1494 genes increases in SCL-II cells and 995 in MET1 overexpressing miR-199a-3p. Of these genes only 395 increases in both cells. Likewise, 1499 genes decrease in SCL-II cells and 1185 in MET1. Of these genes, only 297 decreases in both cells (Appendix A). The EMT-related genes: *SMAD4*, *SERPINE1*, *SERPINE2*, *ID1*, *ID2*, and *COL4A1* decrease in both CSCC lines, similar to the qRT-PCR results. (Appendix A).

Using the miRWalk tool, which combines miRNA-target prediction from 12 different algorithms [28]. We found that from the 297 mRNAs whose expression decreased in both cell-lines by at least 2-fold, 50 genes contained a putative binding site to miR-199a-3p at their 3′UTR by at least 4 out of 12 algorithms (Appendix A). Interestingly, of these 50 genes, 33 were shown to promote EMT, proliferation, or migration of cells.

We further study two putative targets: *SMAD4*, which is a major player in the TGF-β signaling and its involvement in EMT is well known [29], and Fibronectin 1 (*FN1*), which is one of the molecular markers of EMT [30]. We cloned the 3′UTRs of *SMAD4* and *FN1* into psiCHECK-II reporter plasmids. These plasmids were transfected into SCL-II cells that overexpress miR-199a-3p. Although *FN1* and *SMAD4* have at least one putative miR-199a-3p binding site, the luciferase assay failed to suggest that either are biochemical targets of miR-199a-3p (Appendix A).

A second strategy to identify biochemical targets of miR-199a-3p in an unbiased method is the AGO2-PAR-CLIP technique. In this method, anti-AGO antibodies, are used to pull-down complexes of Argonaute-miRNA bound to their targeted mRNAs [31,32]. A total of 4,897,445 reads were mapped to the human genome. Most reads (3,594,003) were annotated to miRNAs sequences and to the 3′UTR of mRNAs (413,146 reads). From these reads, 905 were of miR-199a-3p, and 13,695 reads were mapped to mRNAs harboring miR-199a-3p binding site(s). Of these, 4484 reads had 3′UTR annotation, arising from 50 different mRNAs (Appendix A). This data is summarized in Figure 3A and Appendix A. From these 50 mRNAs, pull-down by AGO, the expression of 12 mRNAs decreased by at least two-fold in one of the two CSCC cell-lines overexpressing miR-199a-3p (Appendix A). Only *AMIGO2* mRNA decreased in both MET1 and SLC-II by more than two-fold in cells overexpressing miR-199a-3p and was pull-down by AGO (Appendix A).

### 2.8. Biochemical Targets of miR-199a-3p

PAR-CLIP provides binding evidence. However, it does not prove regulatory effects. From, the 50 genes identified by the AGO2-PAR-CLIP 12 are known to promote EMT (Appendix A). To test whether these genes are biochemical targets of miR-199a-3p, we cloned their 3′UTRs into psiCHECK-II reporter plasmids. These 12 plasmids were individually transfected into overexpress miR-199a-3p SCL-II cells or control cells. The addition of the 3′UTR of four genes-*AKT*, *VANGL1*, *RAP2B*, and *TGFBR3* onto the reporter plasmid decreased its expression in cells overexpressing miR-199a-3p (Figure 4B), suggesting that their 3′UTR is directly targeted by the overexpressed miRNA. Of these 4 genes, the 3′UTR of *RAP2B* was the most effectively repressed. Interestingly, the *RAP2B* sequence that was pulled down in the PAR-CLIP assay contains a miR-199a-3p binding site that was not predicted in silico. This is probably because this site has only six complementary nucleotides. However, shown in Figure 3C this site is conserved in mammals.

To validate that miR-199a-3p targets *RAP2B* we generated in psiCHECK-WT-RAP2B-3′UTR plasmid a mutation that substitutes the 6 nucleotides CTACTG, complementary to miR-199a-3p seed sequence, with GGGAAA. This substitution abolished the miR-199a-3p effect on the reporter gene expression (Figure 4D mut-RAP2B columns). Moreover, in CSCC cells transfected with the miR-199a-3p mimic, there is a significant decrease of RAP2B protein as shown in (Figure 4E, F). Collectively, these results prove that *RAP2B* is a bona-fide biochemical target of miR-199a-3p.

### 2.9. Effect of miR-199a-3p on RAP2B Signaling

Ras-related protein Rap-2b, encoded by *RAP2B*, is a member of the Ras family of small GTP-binding proteins. It has been shown that Ras tumorigenesis in breast cancer is dependent on FAK signaling [33]. In addition, it was shown in prostate cancer that the effect of RAP2B is mediated at least in part by FAK-phosphorylation [34]. To study whether the effect of miR-199a-3p on RAP2B also affects FAK-signaling, we analyzed its effect on phosphorylated FAK (p-FAK) expression in CSCC cells. Shown in Figure 5A–C in both SCL-II and MET1 cells, miR-199a-3p overexpression results in a 50% reduction of p-FAK while the total expression of FAK is almost unchanged.

Is the inhibition of EMT in CSCC cells overexpressing miR-199a-3p, due in part to altered FAK-signaling? To address this, we treated SCL-II or MET1 cells with PF-573228, a specific inhibitor of FAK-phosphorylation on Tyr(397) [35], and analyzed the effect on EMT-related genes. As shown in Figure 5D, E, in both CSCC cell-lines, PF-573228 treatment results in decreased expression of EMT-related genes.

### 2.10. RAP2B Expression in CSCC Cells and CSCC Tissue

miR-199a-3p expression decreases in CSCC cells and biopsies. Hence, the protein expression of RAP2B should increase in CSCC cells and tissues. Indeed, as shown in Figure 6A,B, in both CSCC cell-lines, the expression of RAP2B is higher than in PHK. In addition, as shown by immunohistochemistry of CSCC FFPE preparations, RAP2B is expressed only in the basal epidermal layer of healthy skin (Figure 6C) but in almost all cells of the tumor (Figure 6D).

## 3. Discussion

Along the stages of transformation from healthy skin to CSCC, we found that miR-199-3p is highly expressed in healthy skin, SE, and AK-KIN1/2 epidermis, but decreases significantly in later stages: KIN3 and CSSC [20]. This finding was confirmed by Wang et al. [24], and by Gillespie et al. [36]. In our screening, miR-199a-3p was one of the most abundant miRNAs in healthy skin biopsies [20]. To understand the expression regulation of miR-199a-3p in CSCC cells/biopsies, two aspects were evaluated. First, miR-199a-3p can be transcribed from two different chromosomal loci. In both the miRNA is located within another gene intron; *DNM2* and *DNM3OS*. Hence, the expression of miR-199a may correlate with the expression of these genes. However, our results disprove this correlation with *DNM2* and support the existence of a negative correlation with *DNM3OS* expression (Figure 1C–F). A similar discrepancy between *DNM30S* and miR-199a expression was found in hepatocellular carcinoma cells upon induction of hypoxia, Zhang L-F et al. demonstrate that hypoxia suppressed miR-199a expression while enhancing *DNM3OS* expression [37]. Second, DNA methylation might play a role in miR-199a-3p expression. In several cancers, the silencing of miR-199a is correlated with DNA methylation of its putative promoter region. In papillary thyroid cancer, both loci are methylated and unmethylated in normal thyroid tissue [38]. Likewise, in ovarian cancer [39]. In testicular tumor malignancy that the promoter region of miR-199a-1 is hypermethylated compared to the healthy tissue, and increased methylation correlates with the malignancy stage [25]. However, this is not the case in all cancers. In both normal brain tissue and in glioblastomas miR-199a-1 promoter is methylated [40]. miR-199a-2 loci is only methylated in normal brain tissue but not in glioblastomas [40]. The methylation of the promoter regions miR-199a-1 or miR-199a-2 in KC or in CSCC is unknown. Indeed, treatment of CSCC cells with 5-AZA enhances miR-199a-3p expression, suggesting that methylation plays a role in its silencing in CSCC (Figure 2A). Interestingly the expression of both *DNM2* and *DNM3OS* were not affected by this treatment (Figure 2B), reinforcing the conclusion that the regulation of miR-199a-3p is not correlated with these two genes. miRNAs are classified as either intragenic when transcribed within protein-coding or non-coding genes. Intergenic miRNAs are further categorized as miRNAs that are transcribed from intronic regions. Intergenic miRNAs, located between genes, are believed to be transcribed from their own promoters [41]. miRNA location is the main factor determining the mechanism of its expression regulation. Liu B. et al. show that from 1881 miRNAs that were in the miRBase data at the time, 918 were transcribed from intronic regions [41]. Baskerville and Bartel found that in most intragenic miRNAs, there is a correlation between miRNAs’ and their host genes’ expression [42]. Obviously, there are exceptions. Liu et al. point to three miRNAs that even show a negative correlation between the host gene and the miRNA expression [41].

Both miR-199a-1 and miR-199a-2 are intronic miRNAs. Data from different cancer types suggest that both are regulated by their promoters and not by the host gene promoter [25,38,39,40]. However, their regulation differs among various cancers and seems to be tissue-dependent.

Although the expression of miR-199a is regulated through methylation, our results suggest that the methylation of the putative promoter regions of both miR-199a-1 and miR-199a-1 does not play a role in this regulation (Figure 2C) in CSCC. These results suggest that an additional factor that regulated the expression of miR-199a but not *DNM2* or *DNM3OS*, and that this factor seems to be silenced by methylation in CSCC cells.

Our results show that miR-199-3p induced MET in CSCC cell-lines. We aimed to identify the biochemical target of miR-199a-3p that mediates its MET-inducing function. To this end, we compared mRNA expression arrays between CSCC overexpressing miR-199a-3p and control CSCC cells. Of the 297 mRNAs that decreased by at least, 2-fold in both cell-lines overexpressing miR-199a-3p, fifty had putative miR-199a-3p binding sites and 33 of the latter are possibly involved in EMT promotion or enhancement of proliferation or migration. These results independently point to the important role of miR-199a-3p in regulating KC proliferation and migration and the transition between mesenchymal to an epithelial phenotype. However, a decrease in specific mRNA expression and in silico prediction of targeting does not prove that the miRNA indeed regulates a given mRNA. Indeed, of the 33-gene list of downregulated putative targets of miR-199a-3p we analyzed two genes; *SMAD4* and *FN1*, which are involved in EMT [29,43] and decreased by more than two-fold in CSCC cells overexpressing miR-199a-3p. The luciferase reporter assays however failed to prove them as biochemical targets of miR-199a-3p, thus implying an indirect effect. Similarly, one of the miR-199a-3p targets that might affect cell migration and proliferation is *CD44*, which is involved in cell migration, tumorigenesis, and metastasis [44]. CD44 was proven as target of miR-199a-3p in osteosarcoma [45] and in CSCC [24].

Other targets of miR-199a that may affect tumor proliferation and metastasis are *ROCK1*, *HIF1α*, *BCAM*, *FZD6*, and *DDR1* [46,47]. However, these targets are targets of miR-199a-5p and not miR-199a-3p.

The second strategy we applied was the Ago2-PAR-CLIP system. Using this technology, we identified 50 mRNAs bound by Ago2 adjacent to a potential binding site for miR-199a-3p (Appendix A). Among them only two genes, *AMIGO2* and *TRAF3*, appeared in both lists of genes, bound by Ago2 and decreased by more than two-fold in both cell-lines overexpressing miR-199a-3p. Nevertheless, of the 50 genes bound to Ago2 and containing miR-199a-3p putative binding sites. Of these genes, 12 which play a role in EMT, cell proliferation or migration were chosen for further study (Appendix A). Although all 12 genes were bound by Ago2 and have putative binding sites for miR-199a-3p, only the 3′UTR of 4 of them had a significant effect on a reporter expression in cells overexpressing miR-199a-3p. Ago2 binding is not an evidence of repression by the miRNA. Indeed, it was shown recently that the assumption that the complex of Argonaute and miRNA bound to the 3′UTR of a protein-coding transcript would lead to repression of gene expression, is not true in many cases [48].

Among the 12 investigated transcripts, *RAP2B*-3′UTR was the most effective in repressing the reporter gene in cells overexpressing miR-199a-3p. We found that the miR-199a-3p seed-complementary sequences in the *RAP2B*-3′UTR, which bound Ago2 and imposed repression on its expression, are located at bases 780–786 of the 3′UTR (Figure 4C). This site is not predicted by bioinformatics algorithm probably because it is considered a “week” 6-mer seed. However, this site is evolutionally conserved in mammals (Figure 4C).

RAP2B is a member of the Ras oncogene family, its function and structure are reviewed by Qu et al. [49]. It promotes migration, proliferation, and invasion in renal carcinoma cells [50], and lung cancer cells [51]. Part of the signaling of RAP2B is through FAK [33]. Hence, inhibition of FAK-phosphorylation should result in inhibition of EMT-related gene expression. The results shown in Figure 5 confirm our hypothesis: The effect of this inhibitor is similar to the effect of miR-199a-3p, which inhibits the expression of RAP2B. RAP2B through FAK-phosphorylation activates the EMT process and inhibition of phosphorylation causes a decrease in EMT-associated gene expression. Moreover, in cells overexpressing miR-199a-3p, there is a significant decrease in FAK phosphorylation, emphasizing that in CSCC, RAP2B through FAK-signaling, induces EMT.

The role of FAK and specifically p-FAK in CSCC cell migration was shown in the FAK knockout model [52]. In addition, increased expression of FAK, and p-FAK was shown in the premalignant and CSCC lesion, as compared to healthy skin [53]. Here, we found that miR-199a-3p regulates FAK activity through regulating RAP2B. Figure 6 shows that RAP2B is expressed in CSCC throughout the tumor cell layers while in healthy skin it is expressed only at the basal layer. The expression of FAK and pFAK showed by Choi et al. is similar to what we found for RAP2B in CSCC lesions [53].

## 4. Materials and Methods

Biopsies used in the study and cells cultures. The samples, skin biopsies, and cultured cells used in this study have been described in our previous study, Formalin fixed paraffin embedded (FFPE) biopsies of 55 cases were obtained from the pathology institute at the Sheba Medical Center, cSCC (*n* = 19), AK KIN3 (*n* = 6), AK KIN1-2 (*n* = 6), SE (*n* = 15) and normal epidermis (*n* = 9) [20]. Primary human keratinocytes (PHK) cells were grown in an 8% CO_2_ incubator at 37 °C, in high Ca^++^ (1.5mM) medium: DMEM/HAM (Biological Industries, Beit Haemek LTD, Beit Haemek, Israel)/10%FBS (Gibco, Thermo Fisher Scientific, Waltham, MA, 02451, USA). Keratinocytes were harvested from mashed skin (left over from plastic surgery) and grown on a feeder layer of 5000 rad γ irradiated 3T3 mice fibroblast cells as described in [54,55]. The study was approved by the institutional review board of the Sheba Medical Center and conducted in adherence to the Declaration of Helsinki protocols number 9776-12-SMC.

### 4.1. Plasmids Cloning

All restriction enzymes used in this study were from (New England Biolabs (NEB), Hitchin, UK). The miR-199a-pre-miRNA oligonucleotides primers (Table 1A) were annealed and were cloned into the HindIII + EcoRI cut pcDNA3.1 (+) plasmid [15,56]. 3′UTRs were cloned into psiCHECK-II vectors (Promega, Madison, WI, 53711, USA). *SMAD4* and *FN1* 3′UTRs were amplified with primers as listed in Table 1B. The amplified PCR fragments were cut with XhoI+NotI and cloned into a psiCHECK-II plasmid that was digested with XhoI + NotI. All other 3′UTRs were amplified by PCR using In-Fusion cloning kit (Clontech Laboratories, Inc. A Takara Bio Company, Mountain View, CA, 95131, USA) and cloned into the PmeI cut site of psiCHECK-II with the In-Fusion cloning kit. Primers used for amplification are shown in Table 1B. To create mut-RAP2B-3′UTR in the miR-199a-3p seed-complimentary region, the Q5^®^ Site-Directed Mutagenesis Kit (New England Biolabs, Ipswich, MA 01938-2723, USA) was used with primers that were designed with the NEBaseChanger tool (Table 1C).

### 4.2. Generation of Stable CSCC Cell Lines

MET1 and SCL-II cell lines were transfected with a plasmid expressing miR-199a or control as described in [57]. The selection was done with G418 antibiotics (Sigma-Aldrich Israel Ltd., an affiliate of Merck, Rehovot, 7670603, Israel). PF-573228 a focal adhesion kinase (FAK) inhibitor (Sigma-Aldrich Israel Ltd., an affiliate of Merck, Rehovot, 7670603, Israel) was used to inhibit FAK phosphorylation.

### 4.3. mRNA Microarray Analysis

mRNA expression microarray analysis was done as described previously by Zehavi et al. [57]. mRNA array hybridization data were deposited in NCBI’s Gene Expression Omnibus, GEO accession number GSE186031.

### 4.4. miRNA Real-Time RT-PCR

Total RNA was purified from cell lines with the Total RNA Purification Kit (Norgen Biotek, Thorold, ON, L2V 4Y6, Canada). miRNAs were quantitated as described previously by Mizrahi et al. [20]. Calculated expression values of miRNAs are presented as ΔΔCT relatively to Rnu43 (SNORD43) or Rnu48 (SNORD48) amplification.

### 4.5. mRNA Real-Time RT-PCR

cDNA was prepared to form total RNA with Takara PrimeScript ™ RT reagent Kit (Clontech Laboratories, Inc. A Takara Bio Company, Mountain View, CA, 95131, USA). Amplification reactions were conducted with Power SYBR^®^ Green Master mix (Applied Biosystems Thermo Fisher Scientific, Waltham, MA, 02451, USA). All primers for RT-PCR were designed using the Primer3 software and are shown in Appendix A. Calculated expression values of studied mRNA were relative to the expression level of the *RPLPO* gene or *GAPDH*.

### 4.6. Western Blot (WB) Analysis

WB analyses were performed as detailed in Mizrahi et al. [20]. The antibodies used in the study are RAP2B (Anti-RAP2B antibody number ab101369) diluted 1:1000, and HSP70 (Antibody (B-6): sc-7298, Santa Cruz Biotechnology Inc., Dallas, TX, 75220, USA) diluted 1:40000. Goat anti-mouse-HRP IgG (Jackson Immuno- Research Lab., West Grove, PA, 19390, USA), or goat anti-rabbit IgG (Sigma-Aldrich Israel Ltd., an affiliate of Merck, Rehovot, 7670603, Israel) both diluted 1:10000, ECL with WESTAR ANTARES (Cyanagen, Bologna, 40138, Italia). Quantitation was performed with the Image Lab program (Bio-Rad Laboratories Ltd. Rishon-Le-Zion, 7565513, Israel), Anti-FAK antibody [EP695Y] (ab40794), Anti-FAK (phospho Y397) antibody [EP2160Y] (ab81298). The WB analyses of EMT markers were performed with the Epithelial-Mesenchymal Transition (EMT) Antibody Sampler Kit #9782 (Cell Signaling Technology, Inc., Danvers, MA, 01923, USA).

### 4.7. Immunohistochemistry (IHC) Staining

Stainings were conducted as described previously by Harari-Steinberg et al. [58]. Paraffin blocks were cut into five µm sections. Sections were pre-treated using OmniPrep solution (pH 9.0) at 95 °C for one hour following the manufacturer protocol (Zytomed Systems, Berlin, 10587, Germany). Next, incubation with Cas-Block solution (Catalog number: 008120) (Thermo Fisher Scientific Waltham, MA, 02451, USA) for 20 min was used for blocking. Then, the slides were incubated O/N 4 °C with anti-RAP2B (Anti-RAP2B antibody number ab101369) diluted 1:500. Detection was done using ImmPRESS™ Anti-Rabbit Ig Reagent Peroxidase (VE-MP-7401) and developed with ImmPACT™ DAB peroxidase (Catalog number: SK-4100) (Vector Laboratories, Inc., Burlingame, CA, 94560, USA).

### 4.8. Real-Time-Cell-Analyzer (RTCA)

xCELLigence DP system (Agilent, Santa Clara, CA, 95051, USA) was used to characterize proliferation, as described previously [15,57,59]. Briefly, 5000–10,000 cells were seeded in E-Plates xCELLigence system tissue culture dish. The E-plates contain micro-electrodes integrated on the bottom which measures the electrical impedance of each bound cell, which is translated into a cell number index. Data were collected continuously for 5 days every 20 min automatically by the analyzer. For migration analysis, CIM-Plates were used. This CIM-plate contains an upper chamber composed of a microporous membrane that cells can migrate through. The analyzer automatically counts each cell that migrates through the porous to the lower E-plates. 5000–10,000 cells were seeded in the upper chamber. Data were collected continuously for 5 days every 20 min automatically by the analyzer.

### 4.9. Ago2 PAR-CLIP, Sequencing and Data Analysis

The assay was performed on SCL-II cells. Cells were grown in the presence of 100 μM 4-thiouridine (4SU) for 24 h and processed as described in Masalha et al. [60]. Ago2 PAR-CLIP data were deposited in NCBI’s Gene Expression Omnibus, GEO accession number GSE155538.

### 4.10. Luciferase Assay for Target Validation

Five hundred thousand SCL-II cells that either stably express mir-199a or empty pcDNA3.1 were seeded in 24-well tissue culture dishes. 24 h later, cells were transfected with 50 ng of reported plasmid using LipofectamineTM 2000 Reagent (Invitrogen- Thermo Fisher Scientific Waltham, MA, 02451, USA) according to the manufacturer’s protocol. Luciferase assay, Dual-Luciferase Reporter (DLR) Assay System (Promega Corporation Madison, WI, 53711, USA) was applied 24 h after transfection according to the manufacturer’s protocol.

### 4.11. Methylation Analysis

Cells were seeded at 50% confluence and were treated with 10 μM 5-AZA-2′deoxycytidine (5-AZA) (Sigma-Aldrich Israel Ltd., an affiliate of Merck, Rehovot, 7670603, Israel) for 5 days (On each day a new medium+ 5-AZA was added). Next, RNA was extracted from the cells and subjected to qRT-PCR of miR-199a-3p or DNM2 and DNM3OS.

### 4.12. Bisulfite Genomic Sequencing

Genomic DNA was extracted from cells or tissue using QIAamp DNA Mini Kit (QIAGEN Sciences Inc, Germantown, MD, 20874, USA). Next, ~1–2 μg of genomic DNA was subjected to bisulfite conversion using the EZ DNA Methylation-Gold Kit (ZYMO research, Irvine, CA, USA).

Next, the bisulfite conversion was subjected to PCR amplification using EpiTaq HS (for bisulfite-treated DNA) (Takara Bio USA, Inc. (TBUSA, formerly known as Clontech Laboratories, Inc.). The PCR fragments were cloned into PUC-19 and cut with BamHI using In-Fusion cloning kit (Clontech Laboratories, Inc. A Takara Bio Company, Mountain View, CA, USA). Plasmid DNA was extracted using GeneJET Plasmid Miniprep Kit (Thermo Fisher Scientific Waltham, MA, USA). At least 5 clones were sequenced from each sample cell line or biopsy.

PCR-specific primers to the DNA converted by the bisulfite were designed using the MethPrimer program [61] (Table 1E).

## 5. Conclusions

This study shows that miR-199a-3p expression decreases in the shift from AK-KIN1/2 to AK-KIN3 and shows that miR-199a-3p plays a major role in controlling the EMT process at least partially through targeting RAP2B in KC. Therefore, miR-199a-3p might be a molecular marker for the state of KC tumor progression. Hence, inhibitors of RAP2B phosphorylation might be potential therapeutic agents in CSCC. Moreover, advanced knowledge of the EMT mechanisms will enhance the development of new therapies for many human tumors, including CSCC, as suggested by Fernandez-Figueras et al. [62].

## Figures and Tables

**Figure 1 ijms-23-15401-f001:**
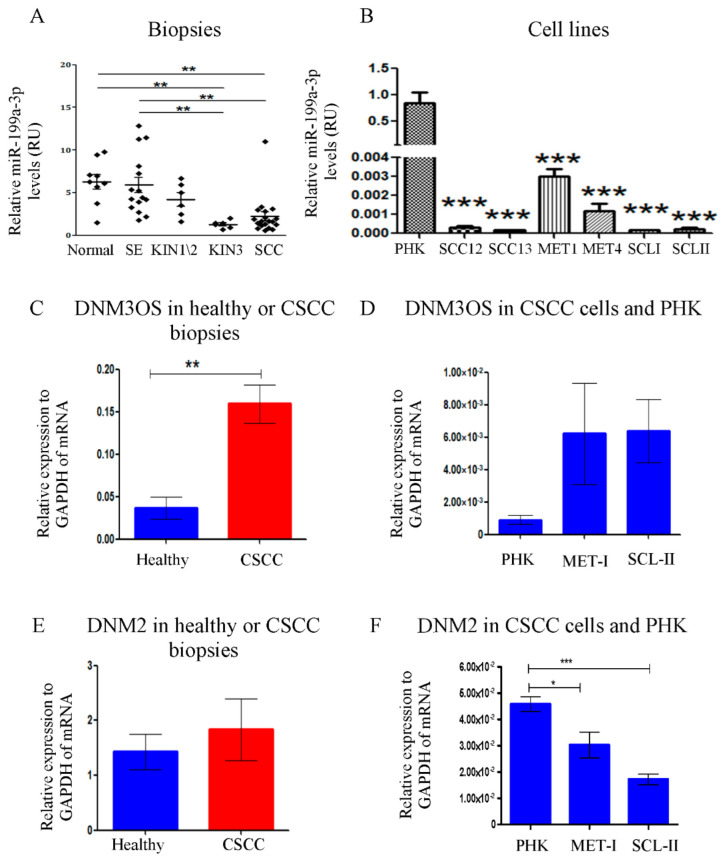
miR-199a-3p expression in different development stages towards CSCC and in CSCC cell-lines. (**A**) miR-199a-3p expression along different stages in the malignant evolution of CSCC. Results are normalized to Rnu43 (SNORD43). Each diamond represents an individual sample. The average is denoted by a horizontal line (the number of samples tested in each stage is written in parentheses below). (**B**) The expression levels of miR-199a-3p in various CSCC cell lines and PHK culture as assessed by qRT-PCR and normalized to Rnu48. (**C**) The expression of DNM3OS in healthy skin biopsies compared to CSCC biopsies (**D**) The expression of DNM3OS in PHK compared to CSCC cell lines. (**E**) The expression of DNM2 in healthy skin biopsies compared to CSCC biopsies. (**F**) The expression of DNM2 in PHK compared to CSCC cell lines. Error bars show ± SEM. *p*-values are calculated via one-way ANOVA, and Tukey’s method was implemented as a correction for multiple comparisons. ** *p* < 0.01, *** *p* < 0.001. *p*-values are calculated via One-Way ANOVA, and Tukey’s method was implemented as a correction for multiple comparisons. * *p* < 0.05, ** *p* < 0.01, *** *p* < 0.001.

**Figure 2 ijms-23-15401-f002:**
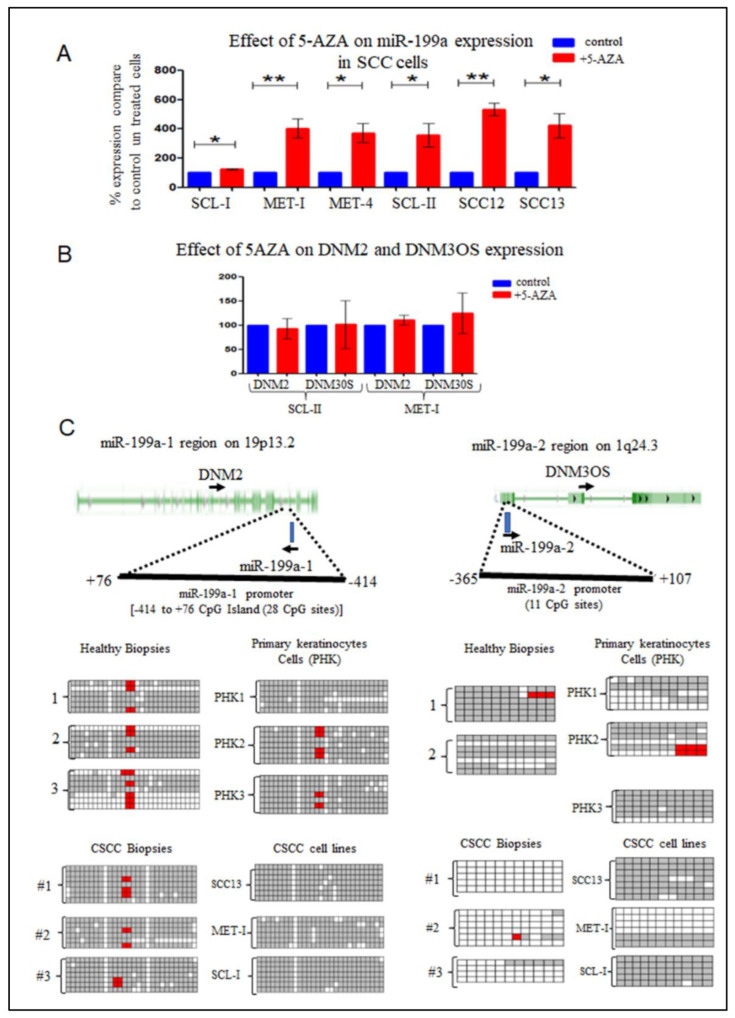
DNA methylation affects miR-199a expression. (**A**) Six CSCC lines were treated with 5-AZA for 5 days. Next, RNA was extracted from both treated and control untreated cells and was subjected to qRT-PCR as written in Figure 1. The mean −/+ SD was calculated from at least 6 independent experiments. Statistics were performed using a *t*-test (* *p* < 0.05) (** *p* < 0.01) (**B**) The effect of treatment with 5-AZA for 5 days on the expression of DNM2 and DNM3OS. (**C**) In the upper panel drawn a scheme of the genomic region of miR-199a-1 and miR-199a-2 genes and the location of the CpG island that was analyzed. Genomic DNA was extracted from three different PHK lines and three different CSCC cell lines or 3 healthy skin biopsies and 3 CSCC biopsies. The genomic DNA was subjected to bisulfite treatment. The treated DNA was subjected to PCR amplification and cloned into PUC19 (see Materials & Methods). From each sample at least 5 sequencings were performed (gray square represents methylated C residues, the white square represents un-methylated C residue, red represents unclear seq of the residue or missing C residue).

**Figure 3 ijms-23-15401-f003:**
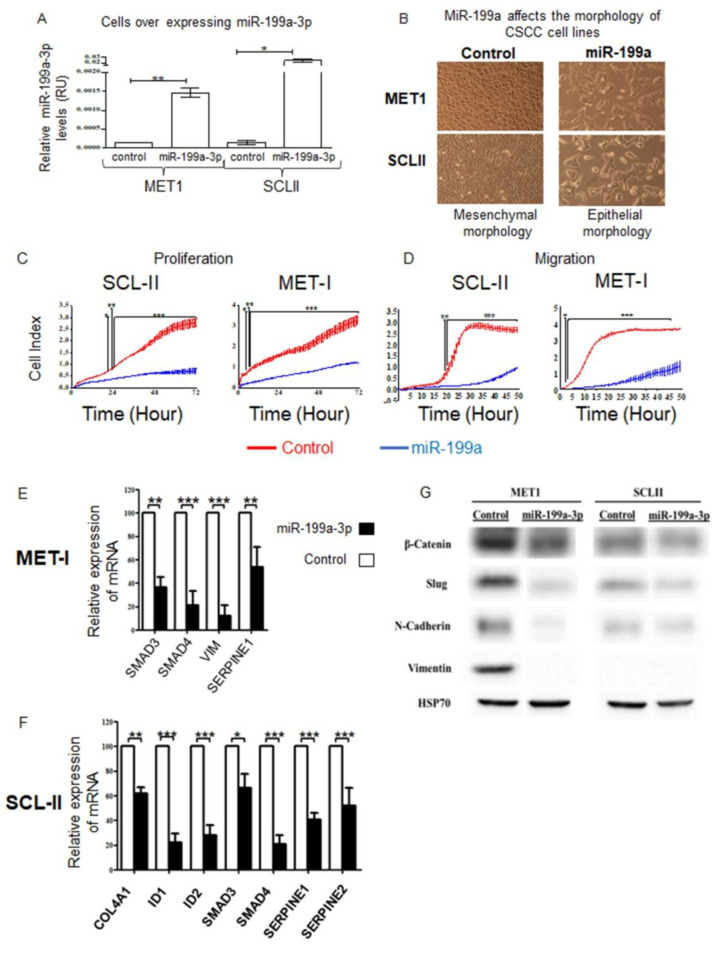
Phenotypic effects of miR-199a-3p on CSCC cells. (**A**) qRT-PCR analysis of miR-199a-3p normalized to Rnu43 in control or MET1 or SCL-II transfected with a plasmid expressing miR-199a-1. (**B**) Morphologic changes were consistent with MET in MET1 and SCL-II cells; the same number of cells was seeded initially in each sample. The photographs were taken under a phase-contrast microscope (magnification ×10). The growth (**C**) and migration (**D**) of MET1 and SCL-II cell-lines stably transfected with the control vector (red) or vector expressing mir-199a (blue) was assessed with xCELLigence TM real-time system. Statistics were performed using two-way ANOVA and Bonferroni post hoc corrections; * *p* < 0.05, ** *p* < 0.01, *** *p* < 0.001. The mRNA levels of EMT-related effectors in cells overexpressing miR-199a-3p. RNA extracts from (**E**) MET1 and (**F**) SCL-II cells were subjected to qRT-PCR with specific primers (see Methods). mRNA expression was calculated relative to RPLP0 expression. The expression of each mRNA, in cells overexpressing miR-199a-3p was relative to its expression in control cells. Values are expressed as the mean + SD of three independent experiments, * *p* < 0.05, ** *p* < 0.01, *** *p* < 0.001. (**G**) Protein extracts from the named cells were subjected to WB analysis with antibodies to the indicated proteins.

**Figure 4 ijms-23-15401-f004:**
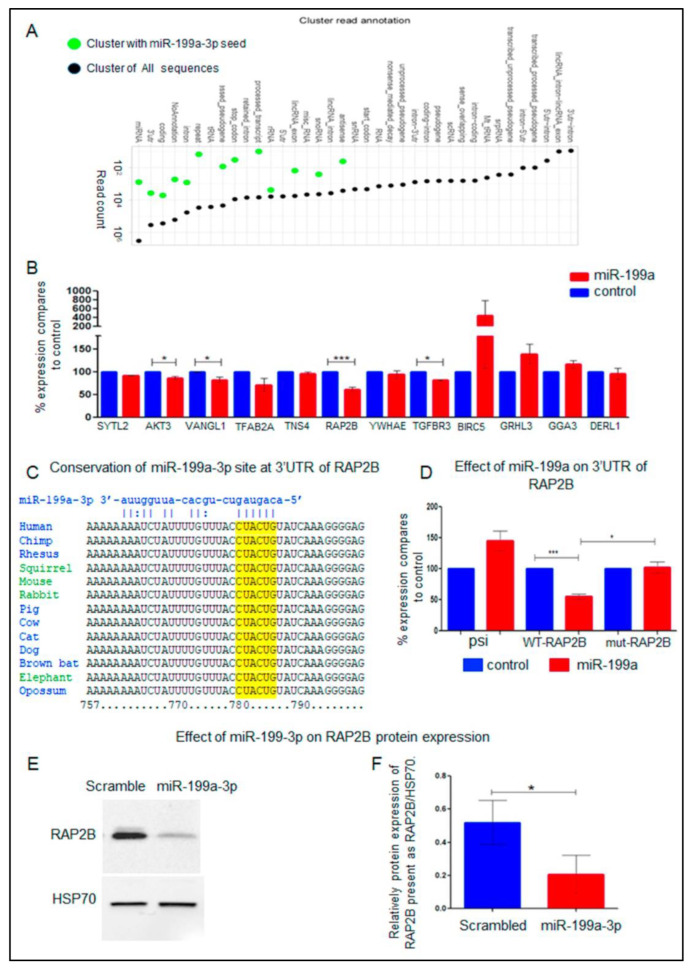
Summary of PAR-CLIP Ago2 assay and RAP2B is a biochemical target of miR-199a-3p. (**A**) Genomic distribution of Argonaute 2 binding sites in transcript regions. Black = all cluster-assigned PAR-CLIP reads; Green = cluster-assigned PAR-CLIP reads containing miR-199a-3p complementary regions. RAP2B is a biochemical target of miR-199a-3p. (**B**) SCL-II cells stably expressing miR-199a (red columns) or control vector (blue columns) were transfected with psiCHECK-II vectors with 3′UTR of the named genes (fused to Renilla luciferase). 24 h after transfection cell lysates were subjected to a dual luciferase assay. The ratio of luminescence of renilla/firefly luciferase obtained in control cells for each gene was set as 100% and the results are the ratio of expression of renilla/firefly luciferase in cells expressing miR-199a-3p relative to controls. The mean ± SD was calculated from 3 independent experiments. Statistics were performed with *t*-tests * *p* < 0.05, *** *p* < 0.001. (**C**) The conservation of a RAP2B 3′UTR region that contains a putative miR-199a-3p binding site and was pulled down in the PAR-CLIP assay is shown in mammals. (**D**) SCL-II cells stably expressing miR-199a-3p (red columns) or control vector (blue columns) were transfected with psiCHECK-II (psi columns) or with a psiCHECK-II vector with WT or mutated 3′UTR of RAP2B fused to renilla luciferase (WT-RAP2B or MUT columns, respectively). The ratio of expression of renilla/firefly control cells was set as 100% and the results are the ratio of expression of renilla/firefly luciferase in cells expressing miR-199a-3p relative to it. The mean ± SD was calculated from 6 independent experiments. Statistics were performed with *t*-tests * *p* < 0.05, *** *p* < 0.001. (**E**) SCL-II cells were transfected with 10 nM of miR-199a-3p mimic RNA or with 10 nM of scrambled control RNA. WB analysis of the RAP2B protein is shown 48 h after transfection. (**F**) Bar plot presenting 3 independent densitometry measurements of the WB experiments shown in D. The data is the relative expression of RAP2B to Heat-Shock Protein 70 (HSP70). Statistics were performed using *t*-tests * *p* < 0.05.

**Figure 5 ijms-23-15401-f005:**
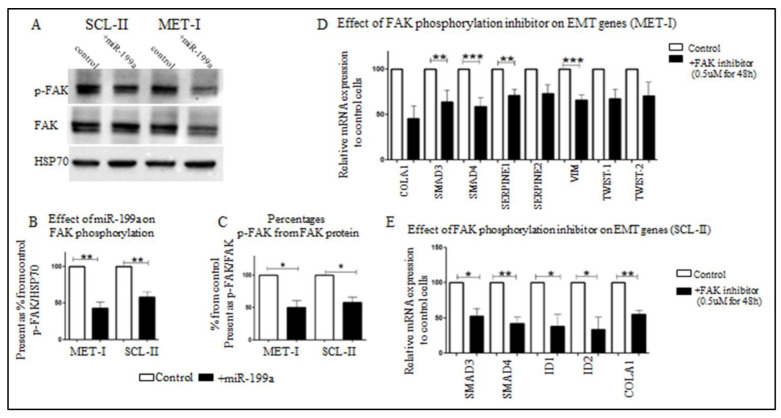
Effect of miR-199a-3p on phosphorylation of FAK and effect of inhibition of FAK-phosphorylation on EMT genes expression in SCC cells. (**A**) WB analysis of protein extracts from SCL-II and MET1 control cells or cells overexpressing miR-199a-3p, as indicated. The extracted proteins were blotted with the indicated antibodies; phosphorylated FAK (p-FAK), total FAK (FAK), and HSP70. p-FAK densitometry is quantified relative to HSP70 (**B**) and total FAK (**C**) in 3 independent WB experiments. Statistics were performed using *t*-tests * *p* < 0.05, ** *p* < 0.01. MET1 (**D**) and SCL-II (**E**) cells were treated with the FAK inhibitor PF-573228, 0.5 µM for 48 h. Next, RNA was extracted from control cells or treated cells and subjected to qRT-PCR with specific primers as indicated (see Methods). Specific mRNA expression was calculated relative to GAPDH. Values are expressed as the mean + SD of at least three independent experiments, * *p* < 0.05, ** *p* < 0.01, *** *p* < 0.001.

**Figure 6 ijms-23-15401-f006:**
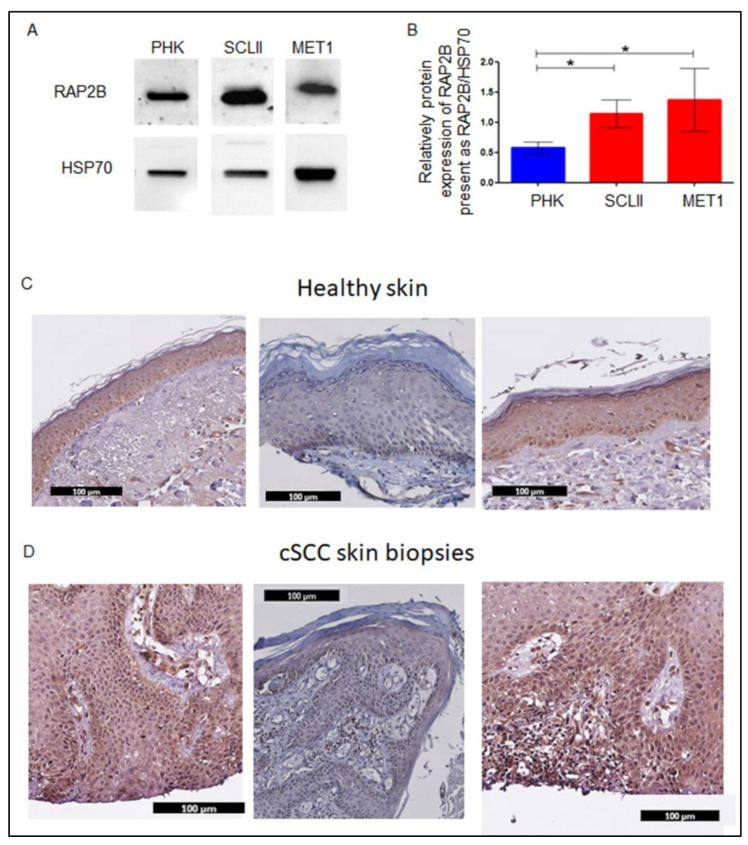
Expression of RAP2B in CSCC cells and CSCC biopsies. (**A**) WB analysis of RAP2B in MET1, SCL-II, and PHK cells, and respective densitometry (**B**). The plot presents 3 independent measurements of WB experiments and shows the relative expression of RAP2B to HSP70. Statistics were performed using *t*-tests * *p* < 0.05. FFPE biopsy slides with mounted normal skin (**C**) and CSCC (**D**) were immune stained with anti-RAP2B antibodies. Antibody binding was visualized with DAB (3,3′-Diaminobenzidine) which yields an insoluble brown shade.

**Table 1 ijms-23-15401-t001:** Primers used in this study.

A: Oligonucleotides Primers for Cloning miR-199a-1 and miR-199a-2				
	Primer-1		Complementary primer	
Pre-mir-199a-1	5′agcttGCCAA**CCCAGTGTTCAGACTACCTGTTC**AGGAGGCTCTCAATGTGT***ACAGTAGTCTGCACATTGGTTA***GGCg-3′		3′aCGGTTGGGTCACAAGTCTGATGGACAAGTCCTCCGAGAGTTACACATGTCATCAGACGTGTAACCAATCCGctta-5′	
Sequences were downloaded from miRBase for miR-199a-1, MI0000242. The bold underline marks the sequence of miR-199a-5p (MIMAT0000231) and the bold italic marks the sequence of miR-199a-3p (MIMAT0000232). The lowercase underline letters marks the nucleotides added to the primers to generate HindIII and EcoRI sites.				
B: Primers used to amplify 3′UTRs				
Gene symbol	NCBI Reference Sequence:	Forward primer	Reverse primer	Legends of PCR product
FN1	NM_002026.3	GTCTCGAGCAGCCAACCAAGATGCAAA	TTTTCCTTTTGCGGCCGCAGGTGGAGGGAAGAAGGGAA	727
SMAD4	NM_005359.5	GTCTCGAGTGGGGCAAGACTGCAAAC	TTTTCCTTTTGCGGCCGCCTCATTCACAGTAAAATGGACCT	877
The added nucleotides to generate XhoI site in the 5′ and NotI of the 3′ PCR product are marked by underline (sequences were added as recommended on the NEB website) https://international.neb.com//media/nebus/files/chartimage/cleavage_olignucleotides_old.pdf?rev=c2f94e1cdcd549c5bf8fdb59f7b63f67&hash=0C83C8F3C59132BE6A5B9A6D4050E3A0.				
RAP2B	NM_002886.3	CCCGGGAATTCGTTTGTGGCTCTTTGCAGCATGTA	GGCCGCTCTAGGTTTCAAATTCATTGCAAGAGATGGA	1033
VANGL1	NG_016548.1	CCCGGGAATTCGTTTGCAGGTGTGTAGCTCAGCAG	GGCCGCTCTAGGTTTCCAGAAGTGCCGAATCATTT	1024
AKT3	NG_029764.1	CCCGGGAATTCGTTTTCCACCCTCTGAGACTCCAT	GGCCGCTCTAGGTTTCCAGCTGGGGCTATTAAAAA	1073
TGFBR3	NG_027757.1	CCCGGGAATTCGTTTGGGCTGAGATTTCCAGGCTA	GGCCGCTCTAGGTTTTTGGAGTTTGGGGCATTTTA	1016
GGA3	NM_138619.3	CCCGGGAATTCGTTTGAACCAAACTGCTGCTGTGA	GGCCGCTCTAGGTTTAGCTAGAGTGGCTGGGACAA	1006
DERL1	NM_024295.5	CCCGGGAATTCGTTTTTCTTGCACACATGCCTCTC	GGCCGCTCTAGGTTTTTGCCTCAAAGTGTGACAGC	1022
SYTL2	NG_029712.1	CCCGGGAATTCGTTTAATGAGCCCAAATTCCACTG	GGCCGCTCTAGGTTTGCCCACTTAGGGGAGATGAT	1066
BIRC5	NG_029069.1	CCCGGGAATTCGTTTCTGGGAAGCTCTGGTTTCAG	GGCCGCTCTAGGTTTAGCATCGAGCCAAGTCATTT	1006
YWHAE	NG_009233.1	CCCGGGAATTCGTTTTTTAGGTTCCTGCCCTGTTG	GGCCGCTCTAGGTTTCTGGAGGACAAGACACACCA	1000
TFAP2A	NG_016151.1	CCCGGGAATTCGTTTGAGCAGGGAAGAGGGTCTTT	GGCCGCTCTAGGTTTCACGGCCTGTTCTGTTCTCT	1002
GRHL3	NG_009308.2	CCCGGGAATTCGTTTCCGTACCCCAAAACAATGTC	GGCCGCTCTAGGTTTGTGCCAACATGACCACACTC	1060
TNS4	NM_032865.5	CCCGGGAATTCGTTTCCCCCTTGCAGATGAGTATC	GGCCGCTCTAGGTTTGCCTGTGACCTTGAGAACCT	
The underlined sequences in both sets of primers are homologous to the 15 nucleotides up and downstream of the PmeI cut site on psiCHECK-II.				
C: Primer used to generate mutation in RAP2B miR-199a-3p binding site				
	Forward primer		Reverse primer	
RAP2B	AAAATCAAAGGGGAGTCTGGG		TCTTGTAAACAAAATAGATTTTTTTTCCACAAATATC	
D: primers used for real-time mRNA real-time RT-PCR				
Gene name	Forward primer		Reverse primer	
DNM2	TACATGCTGCCTCTGGACAA		CTGCTCCGTGTTGAAGATGG	
DNM3OS	AGCCTTCCAGTTTGTACCCT		AGGCAGTTGTGAGCTTAAGT	
SMAD3	ACTACATCGGAGGGGAGGTC		TAGCGCTGGTTACAGTTGGG	
SMAD4	CGCTTTTGTTTGGGTCAACT		CCCAAACATCACCTTCACCT	
SERPINE1	GGGCCATGGAACAAGGATGA		CGGAACAGCCTGAAGAAGT	
SERPINE2	GCAGGACCAAGAAGCAGCTCG		CACGGCGTTAGCCACTGTCACAAT	
COL4A1	GCCCTTCTGCTCCACGAG		CAGTCACATTTGCCACAGCC	
TWIST1	GAGCTGGACTCCAAGATGGC		TCCATCCTCCAGACCGAGAA	
TWIST2	GCAAGAAGTCGAGCGAAGAT		GCTCTGCAGCTCCTCGAA	
ID1	GGCTGTTACTCACGCCTCAA		TGTAGTCGATGACGTGCTGG	
ID2	AGGAAAAACAGCCTGTCGGA		GAGCTTGGAGTAGCAGTCGT	
VIM	GGGAGAAATTGCAGGAGGAG		ATTCCACTTTGCGTTCAAGG	
GAPDH	CTGACTTCAACAGCGACACC		GGTGGTCCAGGGGTCTTACT	
RPLPO	CAGATCCGCATGTCCCTTCG		GCAGCAGTTTCTCCAGAGCTGG	
E. Primers used for amplification of putative promoters regions of miR-199a-1 and miR-199a-2 after DNA was subjected to Bisulfite conversion				
Gene name	Forward primer		Reverse primer	
miR-199a-1	CGGTACCCGGGGATCTGTTATATTTGGAATTGTTTATA		CGACTCTAGAGGATCCAAACCCAACCTAACCAATATACA	
miR-199a-2	CGGTACCCGGGGATCGTTTGAAGATGAAATGATTGTTTAA		CGACTCTAGAGGATCCTCCCTTACCCAATCTAACCAATATA	
The underlined sequences in both sets of primers are homologous to the 15 nucleotides up and downstream of the BamHI cut site on PUC19.				

## Data Availability

The mRNA Microarray Analysis available at GEO accession number GSE186031. The Ago2 PAR-CLIP, Sequencing and Data Analysis available at GEO accession number GSE155538.

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
