# Peer review of "MiR-199a-3p Induces Mesenchymal to Epithelial Transition of Keratinocytes by Targeting RAP2B"

_ijms, 2022, doi:10.3390/ijms232315401_

Round 1
Reviewer 1 Report
The manuscript is well written, the presentation of the work is clear, the conclusions are supported by the evidence presented, and the methods are sufficiently described to allow the study to be repeated, but I have some recommendations before acceptance.
1. The abstract is hard to read. Please rephrase it. It seems like there are fragments of sentences from the main text, and they are hard to understand (lines 25-27, 31-32, 35)
2. Line 77-79; This sentence sounds more like a statement/conclusion than the aim of the study.
3. Please include the p-values in all results sections (e.g., lines 98, 100).
4. Figure 1 is missing the proper legend description below the figure (including all shortcuts)
5. Lines 100-102; should be in the discussion section
6. Lines 118-121; should be in the methods section
7. Lines 123-131; should be in the discussion section
8. Lines 174-188; is it a description of the figure? It is confusing. Lines 179-181 also define the method that was used
9. Line 181; it should be xCELLigence
10. Figures 2 and 3 are missing the proper legend description below the figure (including all shortcuts)
11. Lines 258-281; 296-305; 319-325 please rephrase these sections. Again, it seems to be more of a description of the figure and methods than the results
12. The discussion section is well presented.
13. Lines 434-435; It is worth mentioning how many skin biopsies (with histological types) were obtained. The authors may refer to the detailed description of the material, but it would be beneficial for the reader to mention it.
14. Table 1; different font colors.
Author Response
- The abstract is hard to read. Please rephrase it. It seems like there are fragments of sentences from the main text, and they are hard to understand (lines 25-27, 31-32, 35)
We re-wrote the abstract
- Line 77-79; This sentence sounds more like a statement/conclusion than the aim of the study.
The sentence was moved to the conclusion
- Please include the p-values in all results sections (e.g., lines 98, 100).
- Figure 1 is missing the proper legend description below the figure (including all shortcuts)
The reviewer is right, sections 3, 4, 6, 8, 10, and 11 are figure legends. I don't know if it was my mistake checking the PDF or a mistake in the journal’s technical editing. The endpoint was that in the PDF send to review all the figure legends were inserted as paragraphs in the manuscript body text. In any case, in the revised version, all figure legends were added as part of the figure.
- Lines 100-102; should be in the discussion section
As suggested by the reviewer the sentence was transferred to the discussion
- Lines 118-121; should be in the methods section
- Lines 123-131; should be in the discussion section
As suggested by the reviewer the sentence was transferred to the discussion
- Lines 174-188; is it a description of the figure? It is confusing. Lines 179-181 also define the method that was used
- Line 181; it should be xCELLigence
Thanks for pointing out this typo mistake it was changed
- Figures 2 and 3 are missing the proper legend description below the figure (including all shortcuts)
- Lines 258-281; 296-305; 319-325 please rephrase these sections. Again, it seems to be more of a description of the figure and methods than the results
- The discussion section is well presented.
- Lines 434-435; It is worth mentioning how many skin biopsies (with histological types) were obtained. The authors may refer to the detailed description of the material, but it would be beneficial for the reader to mention it.
A table with a description of the biopsies used in this study was already published by us. Hence, we cannot re-publish the table. However, as the reviewer requested, we added the total number of biopsies and the number of biopsies from each stage.
- Table 1; different font colors.
The colors were omitted from the Table, instead, the nucleosides explanation were marked in bold underlined or bold italic underline, or lowercase underline letters
The manuscript with all corrections is attached

Reviewer 2 Report
Masalha et al. found that miR-199a-3p induces a mesenchymal-o-epithelial transition (MET) in cutaneous squamous cell carcinoma (CSSC) cells. Mechanically, this miRNA inhibits the EMT process by targeting RAP2B and reducing FAK phosphorylation, a downstream biochemical target of RAP2B. A great job, but it needs further major revission as following,
Point 1: Language needs to be polished throughout the manuscript. Sometimes sentences are fragmented. It’s hard to understand.
Point 2: Lines 26-27: It is confused that “We generated CSCC cells overexpressing. In addition, in cells overexpressing miR-199a-3p. We Applied PAR-CLIP assay to search for biochemical targets of miR-199a-3p.” Please rewrite.
Point 3: Inconsistent tenses in the article. Please revise.
Point 4: The order of references in the article is incorrect.
Point 5: Abbreviations should be explained on their first appearance. Such as RAP2B and HSP70.
Point 6: Line 92: “;” is redundant and should be deleted.
Point 7: Line 95: The space before “Hence” is redundant.
Point 8: Line 98: “likewise” shoule be “Likewise”. Carefully correct the errors of the whole MS.
Point 9: “figure” or “Figure”, shold use the same uppercase and lowercase letters in the text.
Point 10: Figure 1: The “miRNA” should be replaced by “miR-199a-3p” in the vertical coordinate title of the Figure 1A and 1B.
Point 11: Each figure or photograph should be provided with a brief description of its content.
Point 12: Lines 108-121: 2.3. CSCC cell-lines section. It seems like the description of figure not the result, and should be moved to the bottom of Figure 1. Please rewrite this section.
Point 13: Lines 123-131: These sentences seem like disscussing about the result. It should be moved to the disscussion section.
Point 14: Lines 165-166: Please rewrite this sentence.
Point 15: Lines 174-188: Move these two paragraphs to the bottom of Figure 3.
Point 16: Lines 254-255: Please show the error bar and significant marker in Figure 3A. Please show the weight of each protein in Figure 3G, Figure 4E in Line 256, Figure 5A in Line 309, and Figure 6A in Line 334.
Point 17: Line 247: “protein WB”, delete “WB”.
Point 18: Lines 296-305: Move them to the bottom of Figure 5.
Point 19: Lines 319-324: Move them to the bottom of Figure 6.
Point 20: Lines 336-431: The discussion section needs further integration.
Point 21: Lines 334: The discussion section needs further integration.
Point 22: Lines 434-443: They should be merged into a single paragraph.
Point 23: Line 444: It should be “Plasmids cloning”.
Point 24: Lines 445-459: They should be merged into one paragraph.
Point 25: Line 481-491 Western blotting section: The dilution of primary and secondary antibodies, substrates, and other reagents should include as well.
Point 26: Line 498: There shold be a space between “4℃with”. And please add the dilution of anti-RAP2B antibody in the IHC assay.
Point 27: Line 508: Please provide more details for migration analysis, including the number of cells, reagent, detect steps, time points and instruments.
Point 28: Supplementary-data Figure-S1: The Gene Ontology (GO) and Kyoto Encyclopedia of Genes and Genomes (KEGG) analysis results of these differentially expressed genes (up-regulated, down-regulated genes) should be added, which could provide the possible regulatory pathway by miR-199a-3p in CSCC, especially whether it contains epithelial-to-mesenchymal transition (EMT) or mesenchymal-o-epithelial transition (MET) pathways.
Point 29: qRT-PCR analysis about several target genes of miR-199a-3p should be provided to verify the reliability of sequencing result.
Point 30: Lines 541 Table 1: Their meaning should be explained below the bottom line of the table as a footnote instead of appearing in the table.
Point 31: Line 546-548: This sentence should be moved to the discussion section.
Point 32: Line 565-711: Provide the doi for references 1-7, 12, 31, 44, 45 and 51, respectively.
Author Response
Point 1: Language needs to be polished throughout the manuscript. Sometimes sentences are fragmented. It’s hard to understand.
We read and check the whole manuscript carefully and change sentences throughout the text.
Point 2: Lines 26-27: It is confusing that “We generated CSCC cells overexpressing. In addition, in cells overexpressing miR-199a-3p. We Applied PAR-CLIP assay to search for biochemical targets of miR-199a-3p.” Please rewrite.
We re-wrote the abstract
Point 3: Inconsistent tenses in the article. Please revise.
Point 4: The order of references in the article is incorrect.
We thank the reviewer for printing out this mistake. The order of the references was corrected and automatically inserted by the Endnote program according to the IJMS format.
Point 5: Abbreviations should be explained on their first appearance. Such as RAP2B and HSP70.
A list of abbreviations explanation is on page number 3 in addition as the review asked, we added an explanation as its first appearance in the text.
I do not know why on the PDF you were asked to review the list was not added.
Point 6: Line 92: “;” is redundant and should be deleted.
Thanks for pointing out this typo mistake obviously it was changed
Point 7: Line 95: The space before “Hence” is redundant.
Thanks for pointing out this typo mistake obviously it was changed
Point 8: Line 98: “likewise” should be “Likewise”. Carefully correct the errors of the whole MS.
Thanks for pointing out this typo mistake obviously it was changed
Point 9: “figure” or “Figure”, should use the same uppercase and lowercase letters in the text.
Thanks for pointing out this to us all figures were changed to be the same Figure with uppercase
Point 10: Figure 1: The “miRNA” should be replaced by “miR-199a-3p” in the vertical coordinate title of the Figures 1A and 1B.
We thank the reviewer for pointing out this. We changed it.
Point 11: Each figure or photograph should be provided with a brief description of its content.
The reviewer is right, points 11, 12, 15, 16, 18, and 19 are figure legends. I don't know if it was my mistake checking the PDF or a mistake in the journal’s technical editing. The endpoint was that in the PDF send to review all the figure legends were inserted as paragraphs in the manuscript body text. In any case, in the revised version, all figure legends were added as part of the figure.
Point 12: Lines 108-121: 2.3. CSCC cell-lines section. It seems like the description of figure not the result and should be moved to the bottom of Figure 1. Please rewrite this section.
Point 13: Lines 123-131: These sentences seem like discussing about the result. It should be moved to the discussion section.
This photograph was moved to the discussion as the reviewer suggested.
Point 14: Lines 165-166: Please rewrite this sentence.
The sentence was rewritten as the reviewer suggested.
Point 15: Lines 174-188: Move these two paragraphs to the bottom of Figure 3.
Point 16: Lines 254-255: Please show the error bar and significant marker in Figure 3A.
Error bars were added to the graph in figure 3A.
Please show the weight of each protein in Figure 3G, Figure 4E in Line 256, Figure 5A in Line 309, and Figure 6A in Line 334.
The WB images were assembled from several experiments. The images for each protein were cut from an image of the gel. Therefore, in my opinion, adding the molecular weight of each protein next to the image is meaningless. We have added a document with the images of all the gels in the supplementary data.
Point 17: Line 247: “protein WB”, delete “WB”.
The sentence was changed
Point 18: Lines 296-305: Move them to the bottom of Figure 5.
Point 19: Lines 319-324: Move them to the bottom of Figure 6.
Point 20: Lines 336-431: The discussion section needs further integration.
Point 21: Lines 334: The discussion section needs further integration.
Point 22: Lines 434-443: They should be merged into a single paragraph.
The paragraph was changed as the reviewer suggested
Point 23: Line 444: It should be “Plasmids cloning”..
The title of the paragraph was changed
Point 24: Lines 445-459: They should be merged into one paragraph.
The paragraph was changed as the reviewer suggested.
Point 25: Line 481-491 Western blotting section: The dilution of primary and secondary antibodies, substrates, and other reagents should include as well.
The dilution of each of the antibodies was added.
Point 26: Line 498: There should be a space between “4℃with”. And please add the dilution of anti-RAP2B antibody in the IHC assay.
The typo mistake was corrected and the dilution of RAP2B in the IHC was added.
Point 27: Line 508: Please provide more details for migration analysis, including the number of cells, reagent, detect steps, time points and instruments.
We added a detailed description of the migration assay
Point 28: Supplementary-data Figure-S1: The Gene Ontology (GO) and Kyoto Encyclopedia of Genes and Genomes (KEGG) analysis results of these differentially expressed genes (up-regulated, down-regulated genes) should be added, which could provide the possible regulatory pathway by miR-199a-3p in CSCC, especially whether it contains epithelial-to-mesenchymal transition (EMT) or mesenchymal-o-epithelial transition (MET) pathways.
We performed both the Functional Annotation Tool DAVID Bioinformatics and The PANTHER (Protein ANalysis THrough Evolutionary Relationships) Classification System GO, for the list of upregulated genes and the down-regulated genes. Beyond the general identification of genes involved in cancer, we did not find a clear characterization of a pathway that changed in the cells. It should be noted that in both of the above analyzes, the EMT pathway is not defined. Hence, we do not think that these analyses suggested by the reviewer would add any important information to the manuscript.
Point 29: qRT-PCR analysis about several target genes of miR-199a-3p should be provided to verify the reliability of sequencing result.
In Figures 3E and 3F present RT-PCR of EMT-related genes that were affected by overexpression of miR-199a-3p.
Point 30: Lines 541 Table 1: Their meaning should be explained below the bottom line of the table as a footnote instead of appearing in the table.
The explanation sentence was moved below the table
Point 31: Line 546-548: This sentence should be moved to the discussion section.
The whole conclusion paragraph in a way is part of the discussion.
Point 32: Line 565-711: Provide the doi for references 1-7, 12, 31, 44, 45 and 51, respectively.
References 1, 5, and 12 do not have doi number. To all other references, the doi number was added.
The manuscript with all corrections is attached

Reviewer 3 Report
This paper clarifies that MiR-199a-3p induces mesenchymal to epithelial transition of keratinocytes by targeting RAP2B. The research is with adequate soundness and originality.
However, in Introduction and Conclusions, the authors should clarify the contribution of this paper more clearly and sufficiently.
Besides, there are some minor format mistakes that need to be corrected. For instance, in Page 15, the font size of sentence "amplified by PCR using In-Fusion cloning kit 452 (Clontech Laboratories, Inc" is larger than common size.
Author Response
However, in Introduction and Conclusions, the authors should clarify the contribution of this paper more clearly and sufficiently.
We read and check the whole manuscript carefully and change sentences throughout the text.
Besides, there are some minor format mistakes that need to be corrected. For instance, in Page 15, the font size of sentence "amplified by PCR using In-Fusion cloning kit 452 (Clontech Laboratories, Inc" is larger than common size.
Thanks to the reviewer for pointing out this typo mistake
The manuscript with all corrections is attached

Round 2
Reviewer 1 Report
The authors addressed all comments.
Reviewer 2 Report
I have no much suggestion on this manuscript.